# Identification and *In Silico* Simulation on Inhibitory Platelet-Activating Factor Acetyl Hydrolase Peptides from Dry-Cured Pork Coppa

**DOI:** 10.3390/foods12061190

**Published:** 2023-03-11

**Authors:** Mingming Li, Xin Zhang, Jiapeng Li, Linggao Liu, Qiujin Zhu, Chao Qu, Yunhan Zhang, Shouwei Wang

**Affiliations:** 1School of Liquor & Food Engineering, Guizhou University/Guizhou Provincial Key Laboratory of Agricultural and Animal Products Storage and Processing, Guiyang 550025, China; 2China Meat Research Center, Beijing 100068, China; 3Beijing Academy of Food Sciences, Beijing 100068, China

**Keywords:** dry-cured pork coppa, bioactive peptides, platelet activating factor acetyl hydrolase (PAF-AH), molecular docking, cardiovascular disease (CVD)

## Abstract

The unique processing technology of dry-cured meat products leads to strong proteolysis, which produces numerous peptides. The purpose of this investigation was the systematic isolation, purification, and identification of potentially cardioprotective bioactive peptides from dry-cured pork coppa during processing. According to the results of anti-platelet-activating factor acetyl hydrolase activity and radical scavenging ability *in vitro*, the inhibitory effect of M1F2 in purified fractions on cardiovascular inflammation was higher than that of M2F2. The peptide of M1F2 was identified by nano-liquid chromatography–tandem mass spectrometry. A total of 30 peptides were identified. Based on bioinformatics methods, including *in silico* analysis and molecular docking, LTDKPFL, VEAPPAKVP, KVPVPAPK, IPVPKK, and PIKRSP were identified as the most promising potential platelet-activating factor acetyl hydrolase inhibitory peptides. Overall, bioactive peptides produced during dry-cured pork coppa processing demonstrate positive effects on human health.

## 1. Introduction

Cardiovascular disease (CVD), including atherosclerosis and hypertension, possesses high morbidity and mortality worldwide [1]. Among them, atherosclerosis is a major potential factor for CVD [2]. Atherosclerosis is a chronic inflammatory process, which might be ameliorated by the confinement of vascular inflammation [3]. Platelet-activating factor acetyl hydrolase (PAF-AH) can hydrolyze low-density lipoproteins (LDLs) to produce pro-inflammatory mediators, including lysophosphatidylcholine (LPC) and oxidized non-esterified fatty acids (oxNEFAs) [4]. Therefore, PAF-AH is considered to be a promising target for CVD treatment [5]. In addition, oxidative stress is also one of the important factors that could promote inflammation by activating the NF-κB pathway, pro-inflammatory gene expression, and inflammatory response [6].

Bioactive peptides that have anti-hypertension, antioxidation, anti-bacterial, and anti-inflammatory functions are composed of 2–20 amino acids [7]. Dry-cured meat products are considered natural sources of bioactive peptides due to their unique processing technology. In previous studies, many bioactive peptides were identified in dry-cured meat products, especially those with antioxidant, anti-ACE, and anti-DPP-IV functions [8,9,10]. With respect to the PAF-AH inhibitory peptides, only a few studies have been reported. Gallego et al. [11] demonstrated the potential of peptides derived from the bone of Spanish dry-cured ham to inhibit PAF-AH activity. In addition, the inhibitory PAF-AH activity of PSNPP, HCNKKYRSEM, and FNMPLTIRITPGSKA, identified from Spanish dry-cured ham, was further confirmed by Gallego et al. [12]. The peptide derived from Panxian dry-cured ham was also confirmed to inhibit PAF-AH activity, where fractions with molecular weight lower than 3 KDa were higher than other fractions [5]. Dry-cured pork coppas are a typical dry-cured meat product, which is pork collar butts that are salted, dried, and ripened [13]. Capocollo di calabria (PDO), coppa piacentina (PDO), and coppa di parma (PGI) are typical dry-cured coppa [14]. In addition, with the globalization of meat consumption markets and the communication of culinary cultures, dry-cured pork coppa has strong market consumption potential in eastern Asian countries, such as China. At present, a large number of studies focus on proteolysis, surface microorganisms, and lipid oxidation in dry-cured pork coppa [13,15,16,17]. However, bioactive peptides from dry-cured pork coppa have been rarely reported. In conclusion, research on anti-PAF-AH peptides in dry-cured meat products requires further exploration and elucidation.

Molecular docking is widely used to investigate the interactions between molecules and predict their binding modes and affinities [18] and was used to investigate not only the structure–activity relationship between macromolecular substrates and protein targets but also the binding affinity of small molecular compounds to biomolecular targets [19]. In the literature on food-derived peptides, much interest is focused on the application of molecular docking to determine the relationship between peptides and angiotensin-converting enzyme and the screening of taste peptides [18,20]. However, few studies have ever focused on evaluating bioactive peptide interactions with PAF-AH using molecular docking.

The aim of this work is to isolate and identify PAF-AH inhibitory peptides from dry-cured pork coppa. Molecular docking was used to further analyze the binding mechanism between dry-cured-pork-coppa-derived peptides and PAF-AH. This investigation could contribute to expanding our understanding of the functional activities of dry-cured-meat-derived peptides and provide a basis for explaining the protective effects of dry-cured-meat-derived peptides in slowing the development of CVD.

## 2. Materials and Methods

### 2.1. Preparation of Peptides

Dry-cured pork coppa preparation followed the method of our previous work [16,21]. Pork collar butts (1.2 ± 0.3 kg) were purchased from the Beijing Dahongmen slaughterhouse. Raw pork collar butts were trimmed to a similar size (28 ± 1.5 cm × 8 ± 1 cm × 6 ± 1 cm) and then salted (3.5% NaCl and 0.01% NaNO_2_, 4 ± 1 °C for 4 days), dried (80% RH, at 8 °C for 10 days), and ripened (65–85% RH, 15 °C for 60 days) in a meat processing plant (Jinmeitian Co., Ltd., Beijing, China). The preparation of peptides was according to a previous procedure [5]. The sample was fractionated by ultrafiltration through a 3 KDa cutoff membrane (Millipore, Shanghai, China), and then the filtrate (< 3 KDa) was vacuum freeze-dried. The ability of PAF-AH inhibitory peptides in different processing stages was evaluated by preliminary experiments. The peptides from the ripening stages (ripening of 34 d as M1 and 44 d as M2) showed inhibitory PAF-AH activity, whereas peptides from other stages did not show inhibitory PAF-AH activity. Therefore, the peptides (M1 and M2) were stored at −80 °C for further analysis.

### 2.2. Analysis of Amino Acids Profile

The extraction of amino acids (AAs) was according to the method of Teka et al. [22] with a slight modification. A 50.0 mg sample and 15 mL HCl (6 mol/L, GR) were mixed, nitrogen-sealed, and hydrolyzed (110 °C for 22 h). The hydrolyzed samples were filtered, and then the filtrate made up a constant volume of 25 mL by the addition of ultrapure water. A 1 mL sample was dried. The dry sample was dissolved by adding 1 mL sodium citrate buffer (pH 2.2) and then filtrated (0.22 μm). According to the description of Gao et al. [23], AA analysis was performed by using an amino acid analyzer (HITACHI L-8900, Tokyo, Japan).

### 2.3. PAF-AH Inhibitory Activity

PAF-AH inhibitory activity was determined by a PAF-AH inhibitor assay kit (Cayman Chemical Co., Ann Arbor, MI, USA). PAF-AH inhibitory activity was calculated according to the following equation:PAF-AH inhibitory activity (%) = (A _blank_ − A _sample_)/(A _blank_ − A _background_) × 100%
where A _blank_ means no sample peptides were added, A _sample_ indicates that the sample peptides were added, and A _background_ is the absorbance of the background reagent.

### 2.4. Fractionation Using Size-Exclusion Chromatography

A G-15 Sephadex column (16 × 60 cm, Cytiva, DC, USA) was used to perform chromatographic fractionation. The sample volume was 1 mL (50 mg/mL, in ultrapure water). The separation was carried out at a flow rate of 1 mL/min with ultrapure water as the eluent at room temperature. The absorbance value was measured at 220 nm. The peptide eluate of 5 mL/tube was collected with an automatic collector, and the same peak components were then mixed. Finally, the peptide eluate was vacuum freeze-dried and stored at −80 °C for further analysis.

### 2.5. Measurement of Radical Scavenging Activity

DPPH· radical scavenging activity was measured according to the previous description with appropriate modifications [5]. Briefly, the mixture solution with 25 μL samples, 125 μL ethanol, and 70 μL DPPH· solution (0.02% in ethanol) was incubated at room temperature in the dark for 60 min. Then, the absorbance value of the mixture was measured at 517 nm. The blank group and background group were prepared by following the same procedure. DPPH· radical scavenging activity was calculated by using the following equation:DPPH · radical scavenging activity (%) = ((A _blank_ − A _background_) − (A _sample_ − A _background_))/(A _blank_ − A _background_) × 100%
where A _blank_ replaces the sample solution with the same solvent, A _background_ replaces the DPPH solution with absolute ethanol, and A _sample_ indicates that the sample peptides were added.

O_2_^−^· radical scavenging activity was evaluated by the previous description of Chen et al. [24] with a slight modification. Briefly, a 100 μL Tris-HCl (40 mM, pH 8.2), 40 μL samples, and 60 μL pyrogallol (3 mM, in 10 mM HCl) were mixed, followed by incubation at 25 °C for 4 min, and absorbance at 325 nm was measured every 30 s. The blank group and background group were prepared according to the above procedure. The blank group replaced the sample with the same solvent, while the background group replaced the sample and pyrogallol with the same solvent. O_2_^−^· radical scavenging activity was calculated by using the following equation:O_2_^−^ · radical scavenging activity (%) = 1 − (K _blank_ − K _sample_)/K _blank_ × 100%
where K _blank_ and K _sample_ are the slopes of the regression equation for the absorbance of the blank group and sample group, respectively.

ABTS^+^ radical scavenging activity was measured by using a commercial assay kit (Suzhou Grace Biotechnology Co., Ltd., Jiangsu, China).

### 2.6. The Analysis of Peptide Sequences by Nano-LC–MS/MS

The samples were separated and analyzed by the UPLC system (Easy-nLC 1200, Thermo Fisher Scientific, Waltham, MA, USA) connected with a C18 column (1.9 μm, 150 μm × 15 cm, 100Å, Dr. Maisch GmbH, Germany). The samples were eluted at a constant rate of 600 nL/min at the following chromatographic conditions. Mobile phase: A: 0.1% formic acid; B: 20% 0.1% formic acid −80% acetonitrile. The linear gradient: from 4% to 8% B for 2 min, from 8% to 28% B for 43 min, from 28% to 40% B for 10 min, from 40% to 95% B for 1 min, and from 95% to 95% B for 10 min. The flow entered directly into the MS/MS system (Q Exactive™ Hybrid Quadrupole-Orbitrap™ Mass Spectrometer, Thermo Fisher Scientific, Waltham, MA, USA) with a 2.1 kV capillary voltage at 270 °C for multiple reaction measurements. MS parameters: MS resolution: 70,000 at 400 m/z; MS precursor m/z range: 300.0–1800.0. MS/MS parameters: Activation type: HCD, normalized coll. Energy: 28.0. Activation time: 66.000 min. Data-dependent MS/MS: up to the top 20 most intense peptide ions from the preview scan in the Orbitrap.

### 2.7. In Silico Analysis of Peptides

An analysis of the potential binding sites for the inhibitory peptides was performed through the Web program PepSite2 (http://pepsite2.russelllab.org/, accessed on 10 January 2023) [25,26]. Potential inhibitory peptides were selected based on their significance level (<5% *p*-value) for binding and the number of potential binding sites.

The human amino acid sequence of PAF-AH was obtained from UniprotKB, and then we used Protein Blast (NCBI, Bethesda, MD, USA) for comparison to obtain the 3D model with the highest homology (PDBID: 3d59) with a high model accuracy (1.5 Å). The 3D model of PAF-AH had water removed and hydrogen atoms added by Discovery Studio 2019 (Accelrys Inc., San Diego, CA, USA).

Toxicity prediction was performed using ToxinPred (https://webs.iiitd.edu.in/raghava/toxinpred/index.html, accessed on 11 January 2023). Web programs (https://web.expasy.org/protparam/ and http://crdd.osdd.net/raghava/hlp/, accessed on 11 January 2023) predicted the stability. ADMET properties were predicted by admetSAR (http://lmmd.ecust.edu.cn/admetsar1/predict/, accessed on 12 January 2023).

### 2.8. Molecular Docking

The peptide molecule was modeled by the Python library (PeptideConstructor) [27], and the all-atom structure was optimized by Discovery Studio 2019 with the charmm force field. Discovery Studio 2019 LibDock module was used to perform molecular docking. The one with the highest LibDockScore was the best docking result.

### 2.9. Statistical Analysis

SPSS 23.0 (SPSS Inc., Armonk, NY, USA) was used for analysis. All data were analyzed using one-factor analysis of variance (ANOVA) and Waller–Duncan test (*p* < 0.05, significant differences). Graphing was performed by using OriginPro 2021(OriginLab Corporation, Northampton, MA, USA) and GraphPad Prism 8 (GraphPad Software Inc., San Diego, CA, USA).

## 3. Results and Discussion

### 3.1. The Separation and Evaluation of Inhibitory PAF-AH Activity and Radical Scavenging Activity of Purified Peptides

Figure 1A illustrates that M1 and M2 had similar anti-PAF-AH activity (5.46% and 5.36%, respectively), which might be caused by the co-regulation of positively charged AAs and nonpolar AAs [28]. M1 and M2 were further purified by a G-15 Sephadex column, and the results are shown in Figure 1(B1,C1). The molecular weight (MW) distributions of M1 and M2 were similar, which were separated into four distinct peaks (F1, F2, F3, and F4). The anti-PAF-AH activity for each fraction was further evaluated. The results are shown in Figure 1(B2,C2). The anti-PAF-AH activity for M1 and M2 considering each fraction was around 7.40–13.73% and 1.90–10.0%, respectively. In previous research, the pre-digestive anti-PAF-AH activity of the Spanish-ham-bone-derived peptides was 5.2–10.4% [11], which is consistent with our findings. Among them, F2 of M1 and M2 showed the highest inhibitory activity (13.73% and 10.0%), which might be attributed to the potential of F2 to contribute more hydrophobic and positively charged AAs related to anti-inflammatory ability. In size-exclusion chromatography, peptides with large MWs were preferentially eluted. F2 was the second eluted fraction with relatively large MWs and a long peptide chain. Similarly, FNMPLTIRITPGSKA containing 15 AAs and TSNRYHSYPWG containing 11 AAs from Spanish ham have higher anti-PAF-AH activities (26.06% and 17.0%, respectively), while the inhibitory activity of the remaining peptides (from pentapeptide to octapeptide) range from 1.28% to 14.0% [12].

Oxidative stress due to the excessive formation of highly reactive radicals is associated with inflammation [6]. Oxidative stress can activate the NF-κB pathway, pro-inflammatory gene expression, and inflammatory response, which induces inflammation [6]. Therefore, the radical scavenging activities of F2 was further evaluated. The radical scavenging activity of M1F2 and M1F2 are shown in Figure 1D. The DPPH·, ABTS^+^, and O_2_^−^· radical scavenging activities of M1F2 were 8.75%, 12.88%, and 14.97%, respectively. In contrast, weak radical scavenging activities were observed in M2F2, which were 6.85% (DPPH·), 9.71% (ABTS^+^), and 11.46% (O_2_^−^·). H_2_O_2_ from O_2_^−^· catalyzed by superoxide dismutase 1 could react with metal cations to generate hydroxyl radicals to promote oxidative stress [29]. Hence, M1F2 and M2F2 have the potential to attenuate the development of inflammation by scavenging radicals to reduce oxidative stress. Compared with M2F2, M1F2 displayed stronger inhibitory PAF-AH activity and radical scavenging activity. Therefore, the amino acid sequences of M1F2 were further identified.

### 3.2. Amino Acids Profile

To understand the contribution of AA composition in M1 and M2 to their functional activity, their AA compositions were further evaluated. As shown in Figure 2, polar AAs were the dominant ones (69.24% and 66.2%), followed by nonpolar AAs (28.59% and 32.17%) in M1 and M2, respectively. The results of the relative ratio of nonpolar amino acids are consistent with those of Xuanwei ham, where the relative ratio of nonpolar AAs was 29% [30]. The function of bioactive peptides was closely related to the AA compositions and their location. In polar AAs, positively charged AAs were abundant (37.39% in M1 and 32.93% in M2), which might be a factor in promoting the anti-inflammatory activity of the dry-cured-pork-coppa-derived peptides, especially at the N- and C-termini [31]. In addition, among nonpolar AAs, Phe was the most abundant amino acid, accounting for 11.3% and 10.76% in M1 and M2, respectively. That has benefits for anti-inflammatory activity [32]. In a previous study, His and Glu were the main AAs of the peptides derived from Jinhua ham and Xuanwei ham [33], which is consistent with the results of our study.

### 3.3. Identification of Peptide Sequences by Nano-LC–MS/MS

Table 1 shows the sequence composition of M1F2. A total of 30 peptides were identified. The prediction by ToxinPred with the platform showed that these peptides were non-toxic. As shown in Figure 3A, these peptides were mainly composed of 5–12 amino acid residues, with the MWs ranging between 500 and 1588.2 Da. Previous work showed that the MWs of the anti-PAF-AH active peptides from Panxian ham had MWs within the range of 248.1–1710.8 Da [5], which is similar to the results of our work. During the processing of dry-cured meat products, the accumulation of bioactive peptides was mainly from muscle proteolysis by endogenous proteases cascades. Figure 3B illustrates that 46.7% and 10% of peptides identified mainly originated from titin and myosin, respectively. Titin and myosin were specific substrate proteins for calpains and cathepsins. VPEVPK, KVPEVPK, VPEVPKK, and VPEVPKKPVP were derived from titin. In addition, the partial amino acid sequences of these peptides were identical. VPEVPK might be generated by the cleavage of K from the N-terminus of KVPEVPK by arginine aminopeptidase (RAP), which was due to the high specificity of RAP for peptides with K at the N-terminus [34]. Similarly, carboxypeptidase B was also highly specific for peptides with K at the C-terminus [35]. As the result, the C-terminal K of VPEVPKK could be released by carboxypeptidase B, resulting in the accumulation of VPEVPK. With respect to VPEVPKK, TPP I might cleave the peptide chain of VPEVPKKPVP to generate VPEVPKK and PVP, which is mainly due to the substrate specificity of hydrolysate of TPP I primarily for tripeptide with hydrophobic AAs at the P1 and P1’ positions [36]. Similarly, TPP I might also play an important role in the accumulation of VPVKKP and DVPVPVKKP derived from myosin. The accumulation of VPVKKP might be attributed to the cleavage of DVP from DVPVPVKKP by TPP I. Therefore, the accumulation of VPEVPK, KVPEVPK, VPEVPKK, VPEVPKKPVP, and DVPVPVKKP was initiated by hydrolyzed titin and myosin via calpains and cathepsins and then modified by TPP I, RAP, and carboxypeptidase B [34,35,36].

Hydrophobic AAs and polar AAs (positively charged AAs) were reported to be present in anti-inflammatory peptides, and many inhibition inflammatory peptides were hydrophobic [5]. The identified peptides contain a high proportion of hydrophobic AAs, such as L, V, P, and A. With respect to positively charged AAs, K was the dominant one. Therefore, LRVFP, IPVPKK, VPEVPK, KVPVPAPK, VEAPPAKVP, PIKRSPD, and so on might be potential anti-inflammatory peptides. Among the identified peptides, KPVGPPNPKP, KVPVPAPK, and KVPEVPK have the same N-termini as the anti-PAF-AH peptides (KAAAAP and KPVAAP) [12]. KVPVPAPK, VPEVPKK, EVPPPKVVK, and VELLK have the same C-terminus as the anti-PAF-AH peptide (NIGK). In addition, peptide N- and C-terminal K residues were associated with the characteristics of the identified heart-healthy peptides [4]. Hence, IPVPKK, KPVGPPNPKP, KVPVPAPK, VPEVPKK, EVPPPKVVK, and so on might be potential anti-PAF-AH peptides from dry-cured pork coppa.

### 3.4. Selection of PAF-AH Inhibitory Peptides Based on Structure–Activity Relationship Analysis

In order to select the peptides with the highest potential to inhibit the PAF-AH activity from the identified peptides, we used the Web program PepSite2 to calculate the intermolecular interactions. As shown in Table 2, a total of 19 peptides met the set criteria among the identified dry-cured-pork-coppa-derived peptides and were considered potential potent inhibitors of PAF-AH. The calculation results of the Web program PepSite 2 showed that the reactive amino acid residues of dry-cured-pork-coppa-derived peptides were dominated by hydrophobic amino acid residues, especially V, L, and P residues. Hydrophobic AAs were the main AAs that make up anti-inflammatory peptides [5]. This might be due to the contribution of hydrophobic amino acid residues to the reactive sites of anti-inflammatory peptides. In addition, positively charged AAs were also one of the contributors to anti-inflammatory peptides. In this investigation, the K residues were the major reactive site contributors among positively charged AAs. With respect to PAF-AH, a range of 3–13 amino acid residues have an interaction with these peptides. Among these peptides, LTDKPFL, PIKRSP, VEAPPAKVP, IPVPKK, and KVPVPAPK were considered the most promising potential inhibitors based on their significance level for binding and the number of potential binding sites. LTDKPFL and PIKRSP bound the highest number of reactive sites (13), followed by VEAPPAKVP, IPVPKK, and KVPVPAPK, which bound 11, 11, and 10 reactive sites of PAF-AH, respectively. Among the amino acid residues in reaction sites, Ser273 and His351 were the amino acid residues that constitute the catalytic triad in the reactive center of PAF-AH, while Leu153 was the amino acid residue that constitutes the oxyanion hole [37]. These potential PAF-AH inhibitory peptides might inhibit PAF-AH activity by interacting with these amino acid residues.

The drug-oriented properties, such as stability and ADMET properties, etc., are important indicators for further evaluation of bioactive peptides [38]. The drug-directed properties of five peptides were further predicted *in silico*, which are presented in Table 3. LTDKPFL was the most stable peptide, and the instability index was 13.19. Theoretically, the stability *in vivo* affects its biological activity [38]. Thus, LTDKPFL might possess much stronger PAF-AH inhibitory activity than PIKRSP, VEAPPAKVP, IPVPKK, and KVPVPAPK. Human intangible absorption (HIA) was one of the important indicators for evaluating the degree of intestinal absorption. As illustrated in Table 2, the intestinal tract could absorb LTDKPFL, IPVPKK, and PIKRSP well. Cytochrome P450s enzymes are important degradative enzymes in liver metabolism, and inhibitors of cytochrome P450s enzymes could lead to drug accumulation poisoning [39]. Cytochrome P4503A4 is one of the important components of cytochrome P450s [39]. The prediction results showed that LTDKPFL, PIKRSP, VEAPPAKVP, IPVPKK, and KVPVPAPK were not inhibitors of cytochrome P4503A4, which means that these peptides have no risk of liver toxicity. Moreover, LTDKPFL, PIKRSP, VEAPPAKVP, IPVPKK, and KVPVPAPK were also non-toxic in the ToxinPred predictions (Table 1). The drug-directed properties showed that these peptides have good metabolic characteristics and were safe and worthy of further study.

### 3.5. Molecular Docking of the Peptides with PAF-AH Enzyme

The docking results of PAF-AH and selected peptide molecules are shown in Table 3 and Figure 4. These peptides have high LibDockScores and rapidly attach to the reactive site of PAF-AH. Ser273, His351, and Asp296 together constitute the catalytic triad of the PAF-AH active site [37]. Leu153 and Phe274 could construct an oxyanion hole, which stabilizes the negative charge of the tetrahedral intermediate via amide nitrogen [40]. LTDKPFL formed 11 hydrogen bonds with amino acid residues in the reactive site of PAF-AH (Figure 4A). Among them, Ser273 and Leu153 related to LTDKPFL through three hydrogen bonds, and the corresponding distances were 2.36 Å, 2.30 Å, and 1.07 Å, respectively. In addition, a Pi–Alkyl interaction (4.04 Å) occurs with His351. The LibDockScore of VEAPPAKVP and PAF-AH was 175.773. PAF-AH interacts with VEAPPAKVP via 10 hydrogen bonds (Figure 4B). Ser273, Leu153, and His351 contributed two, two, and one hydrogen bonds, respectively. It was worth noting that there was also a salt bridge between VEAPPAKVP and His351, which might facilitate its ability to inhibit PAF-AH activity. Previous reports showed that salt bridge formation between inhibitory peptides and PAF-AH might be its pathway for inhibiting PAF-AH [4]. Similarly, the LibDockScore of KVPVPAPK and PAF-AH was 170.304, forming 10 hydrogen bonds (Figure 4C). Ser273, Leu153, Phe274, and His351 were bound to the KVPVPAPK by two, two, one, and one hydrogen bonds, respectively. Moreover, there was an attractive charge (5.09 Å) interaction between KVPVPAPK and His351. Diisopropyl fluorophosphate, an inhibitor of PAF-AH, could modify the Ser273, resulting in changes in the activity of PAF-AH [41]. Hence, interaction with Ser273 might be one of the molecular pathways to inhibit PAF-AH activity. In this investigation, LTDKPFL, VEAPPAKVP, and KVPVPAPK had the potential for anti-PAF-AH activity, which might be attributed to the formation of hydrogen bonds between these peptides and Ser273 of the reactive center on PAF-AH. The LibDockScores of IPVPKK and PIKRSP to PAF-AH were 147.085 and 130.936, respectively (Figure 4D,E). They interacted with PAF-AH via six and four hydrogen bonds, respectively. Previous reports have shown that Met117 of PAF-AH had proximity to the LDL-binding domain [37]. Therefore, LTDKPFL and VEAPPAKVP possess the potential ability to inhibit PAF-AH activity, which could also be attributed to the existence of van der Waals and hydrogen bond (2.92 Å) between LTDKPFL and VEAPPAKVP and Met117, resulting in the association of PAF-AH and LDL being influenced. All things considered, LTDKPFL, VEAPPAKVP, KVPVPAPK, IPVPKK, and PIKRSP could interact with amino acid residues in the reactive center on PAF-AH, such as hydrogen bonds, salt bridge, attractive charge, van der Waals, and so on, thereby inhibiting PAF-AH activity.

## 4. Conclusions

Dry-cured pork coppa was an important source of bioactive peptides. The results of this study indicated that anti-PAF-AH peptides were formed during dry-cured pork coppa processing. The anti-PAF-AH activities of M1 and M2 were 5.46% and 5.36%, respectively. After G-15 Sephadex column purification, M1F2 showed a much stronger ability of anti-PAF-AH activity and radical scavenging activity than M2F2. The fraction M1F2 was identified by nano-LC-MS/MS with 30 peptides. Based on bioinformatics methods, including *in silico* analysis and molecular docking, LTDKPFL, VEAPPAKVP, KVPVPAPK, IPVPKK, and PIKRSP were identified as the most promising potential PAF-AH inhibitory peptides. These peptides could interact with amino acid residues in the reactive center of PAF-AH through hydrogen bonding, charge attraction, van der Waals, and so on. This study expands our understanding of the functional activity of dry-cured-meat-product-derived peptides and provides evidence for their protective effect in slowing the aggravation of CVD. Future studies are required to further investigate the functional activity of these peptides both *in vivo* and *in vitro*, as well as to gain deeper insight into the molecular mechanism of the anti-PAF-AH activity.

## Figures and Tables

**Figure 1 foods-12-01190-f001:**
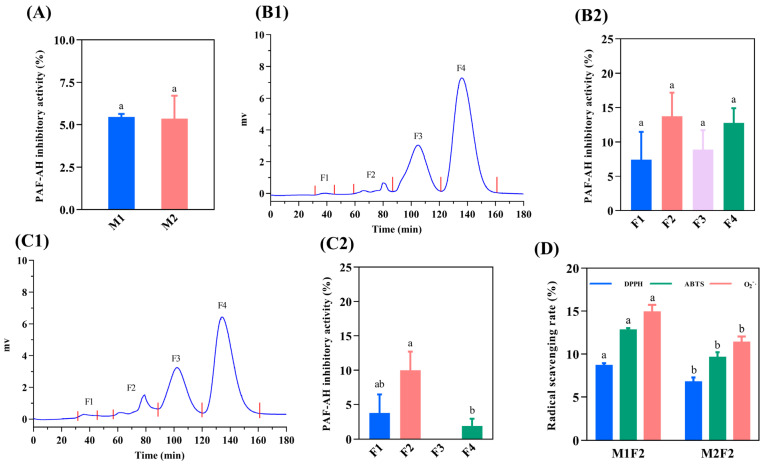
Isolation of peptides derived from dry-cured pork coppa by G-15 chromatography and function evaluation. The PAF-AH inhibitory activity of <3 KDa peptides (**A**). G-15 size-exclusion chromatography of peptides from M1 (**B1**) and M2 (**C1**). The PAF-AH inhibitory activity of purified fractions from M1 (**B2**) and M2 (**C2**). The radical scavenging activity of F2 fractions (**D**). The function evaluation was performed under a peptide concentration of 5 mg/mL. (**B2**,**C2**): Different lowercase letters indicate a significant difference, *p* < 0.05. (**D**): Different lowercase letters indicate a significant difference between M1F2 and M2F2, *p* < 0.05.

**Figure 2 foods-12-01190-f002:**
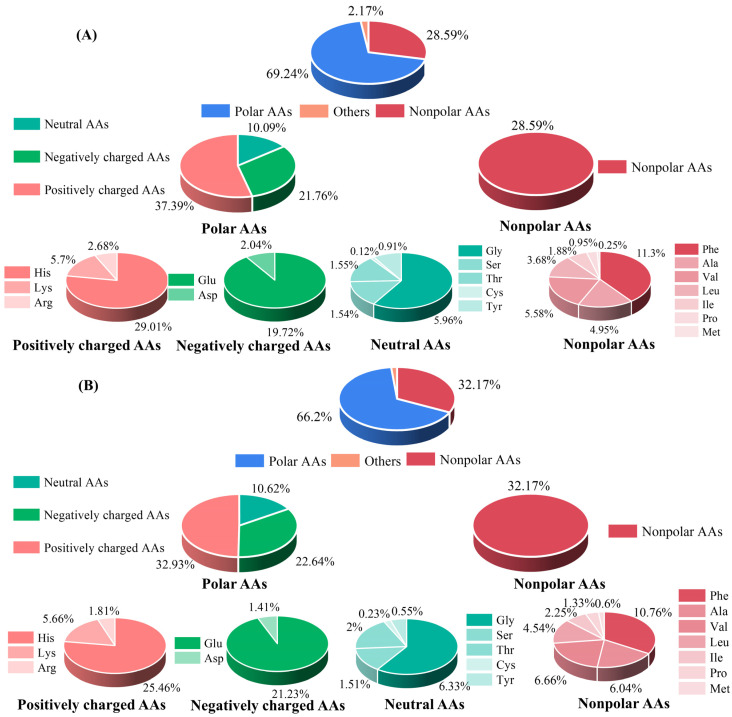
Amino acids profile of M1 (**A**) and M2 (**B**).

**Figure 3 foods-12-01190-f003:**
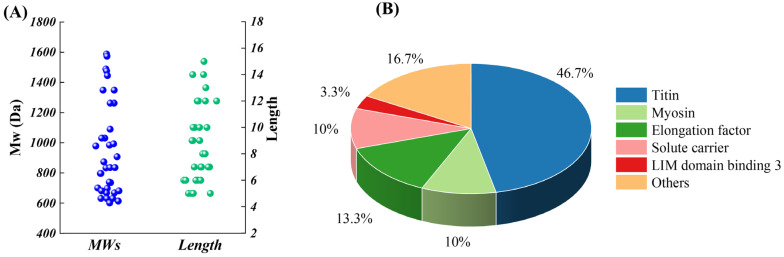
Distribution of MW, length, and source protein percentage of the identified peptide. (**A**) MWs and length; (**B**) source protein percentage.

**Figure 4 foods-12-01190-f004:**
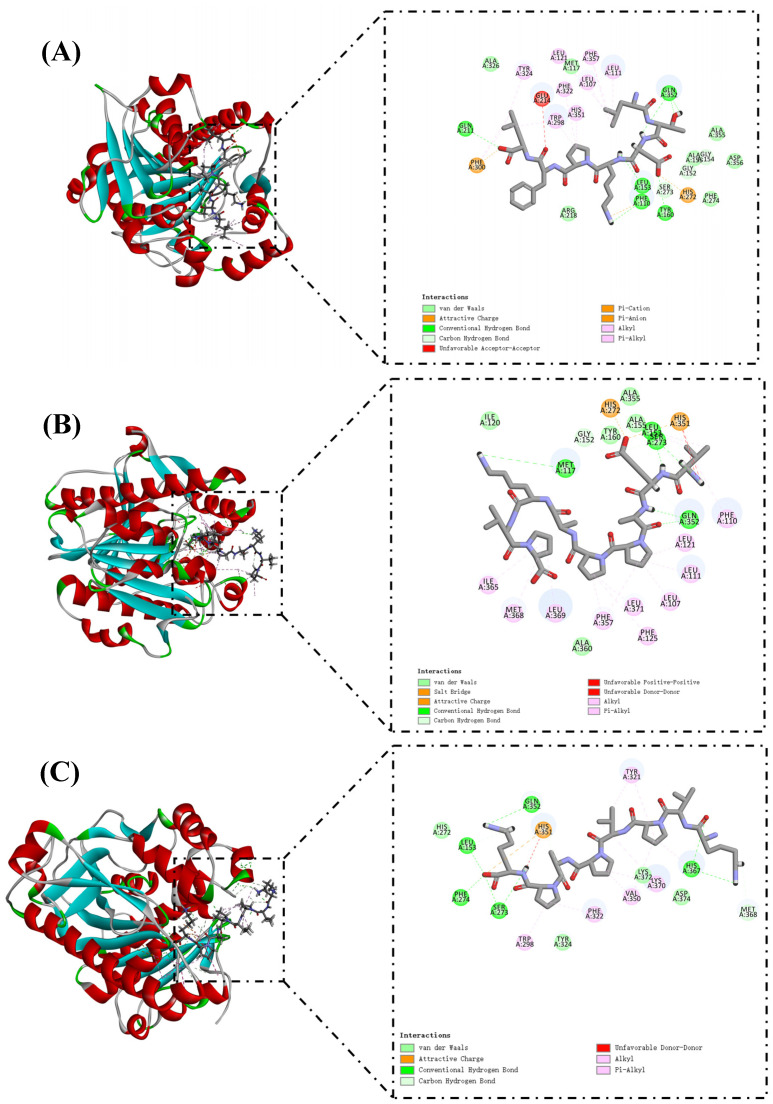
Molecular docking of peptides LTDKPFL (**A**), VEAPPAKVP (**B**), KVPVPAPK (**C**), IPVPKK (**D**), and PIKRSP (**E**) with PAF-AH.

**Table 1 foods-12-01190-t001:** Peptide sequences identified for the MIPF2 fraction in dry-cured pork coppa.

Peptide	Length	MWs	Source Protein	Toxin
LRVFP	5	631.391	ATP synthase mitochondrial	non-toxin
ILLVR	5	613.437	Solute carrier	non-toxin
VELLK	5	601.394	Solute carrier	non-toxin
TLIKR	5	630.430	Receptor protein-tyrosine kinase	non-toxin
IPVPKK	6	681.463	Titin	non-toxin
VPEVPK	6	668.396	Titin	non-toxin
VVELLK	6	700.460	Solute carrier	non-toxin
PIKRSP	6	697.434	SH3 domain	non-toxin
VPVKKP	6	667.448	Myosin	non-toxin
LTDKPFL	7	833.479	Elongation factor Tu	non-toxin
YIPEPVR	7	873.484	Elongation factor Tu	non-toxin
VPEVPKK	7	796.496	Titin	non-toxin
VPEVPKA	7	739.433	Titin	non-toxin
VPPVVPK	7	735.476	Titin	non-toxin
KVPEVPK	7	796.490	Titin	non-toxin
KVPVPAPK	8	835.538	Titin	non-toxin
TIGHVDHG	8	835.406	Elongation factor Tu	non-toxin
VEAPPAKVP	9	907.531	Titin	non-toxin
EVPPPKVVK	9	992.612	Titin	non-toxin
DVPVPVKKP	9	978.602	Myosin	non-toxin
KPVGPPNPKP	10	1030.599	Myosin	non-toxin
VTLTGPGPWG	10	984.523	LIM domain binding 3	non-toxin
VPEVPKKPVP	10	1089.672	Titin	non-toxin
VPEEKVPVPVQK	12	1348.789	Titin	non-toxin
EVPPVTVPEAPK	12	1262.700	Titin	non-toxin
PKPRKHLKPEQS	12	1444.853	Spen family transcriptional repressor	non-toxin
VPEVPKKPVPEEK	13	1475.852	Titin	non-toxin
VEAPPAKVPEVPKK	14	1488.873	Titin	non-toxin
LLLLLLLLLLLLVL	14	1588.203	ADAM metallopeptidase	non-toxin
DFPGDDTPIIIGSAR	15	1573.7846	Elongation factor Tu	non-toxin

**Table 2 foods-12-01190-t002:** Dry-cured-pork-coppa-derived peptides and their binding potential with PAF-AH as protein receptor.

Peptide	*p*-Value	Reactive Residues in Peptide	Bound Residues of PAF-AH
KPVGPPNPKP	0.002511	P5, P6, N7, P8	Phe110, Leu153, Trp298, Phe322, Tyr324
LTDKPFL	0.004142	T2, D3, K4, P5, F6	Gly152, Leu153, Ser273, Phe274, Trp298, Met299, Phe300, Pro301, Phe322, Tyr324, Asn327, Lys330, His351
VEAPPAKVP	0.001218	A3, P4, P5, A6, K7	Gly152, Leu153, Gln211, Glu214, Arg218, Ser273, Phe274, Trp298, Phe322, Tyr324, His351
YIPEPVR	0.005236	Y1, I2, P3, E4, P5	Phe110, Gly152, Leu153, Ser273, Phe274, Trp298, Phe322, Tyr324, His351
KVPVPAPK	0.002309	V4, P5, A6, P7, K8	Phe110, Leu153, Gln211, Glu214, Arg218, Ser273, Trp298, Phe322, Tyr324, His351
VPEVPKK	0.00509	P2, E3, V4, P5, K6	Phe110, Leu153, Gln211, Glu214, Arg218, Trp298, Phe322, Tyr324
VPEVPKKPVP	0.01093	K6, K7, P8, V9, P10	Phe110, Leu153, Gln211, Glu214, Arg218, Trp298, Phe322, Tyr324
LRVFP	0.001057	L1, V3, F4, P5	Trp298, Met299, Phe300, Pro301, Phe322, Tyr324, Ala326, Asn327, Lys330
EVPPPKVVK	0.0004336	V2, P2, P3, P4, K5	Phe110, Leu153, Gln211, Glu214, Arg218, Trp298, Phe322, Tyr324
ILLVR	0.0275	I1, L2, L3, V4	Phe110, Gly152, Leu153, Ser273, Phe274, Trp298, Phe322, Tyr324, His351
IPVPKK	0.0003817	I1, P2, V3, P4, K5	Gly152, Leu153, Gln211, Glu214, Arg218, Ser273, Phe274, Trp298, Phe322, Tyr324, His351
VPEVPK	0.00137	P2, E3, V4, P5, K6	Phe110, Leu153, Gln211, Glu214, Arg218, Trp298, Phe322, Tyr324
DVPVPVKKP	0.00382	V1, E2, L3, L4	Phe110, Leu153, Gln211, Glu214, Arg218, Trp298, Phe322, Tyr324
VPEVPKA	0.00509	P2, E3, V4, P5, K6	Phe110, Leu153, Gln211, Glu214, Arg218, Trp298, Phe322, Tyr324
TLIKR	0.01366	L2, I3, K4, R5	Trp105, Trp115, Asn119
VPPVVPK	0.003036	V1, P2, P3, V4	Phe110, Leu153, Trp298, Phe322, Tyr324
PIKRSP	0.005136	I2, K3, R4, S5, P6	Phe110, Leu153, Ser273, Trp298, Met299, Phe300, Pro301, Phe322, Tyr324, Ala326, Asn327, Lys330, His351
VPVKKP	0.00365	V1, P2, V3, K4, K5	Phe110, Leu153, Glu214, Arg218, Ser273, Trp298, Phe322, Tyr324, His351
KVPEVPK	0.00509	P3, E4, V5, P6, K7	Phe110, Leu153, Gln211, Glu214, Arg218, Trp298, Phe322, Tyr324

**Table 3 foods-12-01190-t003:** In silico prediction of properties of the identified PAF-AH inhibitory peptides.

Peptide	HIA	CYP450 3A4 Inhibitor	Instability Index	Stability	LibDockScore
LTDKPFL	0.7091 (+)	0.8987 (−)	13.19	High	180.22
VEAPPAKVP	0.6421 (−)	0.9097 (−)	85.06	Normal	175.773
KVPVPAPK	0.6337 (−)	0.8549 (−)	118.51	Normal	170.304
IPVPKK	0.7832 (+)	0.8812 (−)	67.73	Normal	147.085
PIKRSP	0.7065 (+)	0.8855 (−)	194.98	Normal	130.936

## Data Availability

Data was contained within the article.

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
