# Peer review of "Identification and In Silico Simulation on Inhibitory Platelet-Activating Factor Acetyl Hydrolase Peptides from Dry-Cured Pork Coppa"

_foods, 2023, doi:10.3390/foods12061190_

Round 1
Reviewer 1 Report
The article brings enough data to present and discuss the topic researched and is adequate as to the need for future studies on of these peptides both in vivo and in vitro, as well as deeper insight into the molecular mechanism of the anti-PAF-AH activity.
Author Response
请参阅附件

Reviewer 2 Report
General comment.
This paper is about the PAF-AH inhibitory ability of peptides extracted from dry-cured pork coppa. This is in line with the journal's purpose in that it illustrated how to analyze peptide properties after extracting the peptides from dry-cured pork coppa, not just the structural docking between peptides and PAF-AH. In addition, tables and figures were properly presented throughout the paper, making it easy to understand the detailed explanations of the experiments. However, there were some errors and misspellings need to be corrected, so I would like to suggest a revision of the detailed part of the paper. Below, revision and questions which I would like to ask while reviewing the paper were presented. With this correction, I hope that this paper will get better quality.
Line 30 : lso-phosphatidylcholine → lyso-phosphatidylcholine
Line 36-47 : In paragraph 2 of the introduction, it would be better to add the explanation about dry-cured pork coppa. And suggesting the reason for using pork coppa in the experiment would make this paper better quality.
Line 67 : PAH-AF → PAF-AH
Line 64-71 : In 2.1 preparation of peptides, dry-cured pork coppa preparation has to be described in more detail. The referenced paper presented 4 way that substitute sodium with various metal ions to cure the pork. It needs to be mentioned more accurately which method was used to prepare the dry-cured pork coppa in this paper.
Line 133 : I suggest that check the address of Web program Pepsite2. When I click on that address, I got an error visiting the site.
Line 136-139 : The sentences need to be grammatically or semantically modified a little.
Line 163 : anti-PAH-AH → anti-PAF-AH
Line 218 : It would be better to add a meaning of abbreviation, RAP.
Fig 2(A) : The numerical value of the polar AAs in the top graph and negatively charged AAs in the bottom graph do not match respectively. The ratio of negatively charged AAs in green colored part drawn in middle seems to be misrepresented.
Reviewer 3 Report
The manuscript entitled "Identification and in silico simulation on inhibitory PAF-AH peptides from dry-cured pork coppa" submitted by Li et al., represents a good contribution to the field of bioactive peptides from dry-cured meat products. However, a major revision is required.
1) Please, do not use abbreviations in title and Abstract section (some examples: PAF-AH, LC-MS/MS and others)
2) It's really hard to understand from the actual version of the manuscript the rationale related to the utilisation of dry-cured pork coppa. What is the reason? Please, provide more information regarding sample collection and more details regarding the product. Is it a commercial product? What is the ripening time of the product? Is it a certified coppa product? Please, consider that in the European contest there there are several certified dry-cured pork coppa samples (such as Coppa Piacentina PDO and others). Take a look and refer to some recent publications on this topic.
3) The M&M related to extraction and purification of peptides and amino acids is well carried out and described.
4) The assays used to evaluated the in vitro antioxidant activity of the peptides are quite banned from high-impact journal (DPPH, ABTS, ORAC, FRAP, and others). Please, describe advantages and drawbacks of these in vitro spectrophotometric assays.
5) More methodological and instrumental information must be provided for the nano-LC-MS/MS to better understand what type of MS equipment was used together with the parameters correlated to the MS-separation.
6) Molecular docking and in silico prediction is well carried out and represents a very novel contribution to the field.
7) According to this reviewer, the Results and Discussion section must be improved. It seems a short communication in this form, not a Research Article.
8) Please, re-organize the Figures; in this format, several figures are not easy to understand (such as the Figure 2, Figure 3, Figure 4). I suggest to improve the resolution of the figures together with the size of the font.
Author Response
请参阅附件

Round 2
Reviewer 3 Report
The manuscript has been accurately revised and can be published in this form.